# Brain Renin–Angiotensin System as Novel and Potential Therapeutic Target for Alzheimer’s Disease

**DOI:** 10.3390/ijms221810139

**Published:** 2021-09-20

**Authors:** Raúl Loera-Valencia, Francesca Eroli, Sara Garcia-Ptacek, Silvia Maioli

**Affiliations:** 1Center for Alzheimer Research, Department of Neurobiology Care Sciences and Society, Division of Neurogeriatrics, Karolinska Institutet, 171 64 Stockholm, Sweden; francesca.eroli@ki.se; 2Center for Alzheimer Research, Department of Neurobiology Care Sciences and Society, Division of Clinical Geriatrics, Karolinska Institutet, 171 64 Stockholm, Sweden; sara.garcia-ptacek@ki.se; 3Aging and Inflammation Theme, Aging Brain Theme, Karolinska University Hospital, 141 57 Stockholm, Sweden

**Keywords:** Alzheimer’s Disease, renin-angiotensin system, mouse models, cognition, hypertension

## Abstract

The activation of the brain renin-angiotensin system (RAS) plays a pivotal role in the pathophysiology of cognition. While the brain RAS has been studied before in the context of hypertension, little is known about its role and regulation in relation to neuronal function and its modulation. Adequate blood flow to the brain as well as proper clearing of metabolic byproducts become crucial in the presence of neurodegenerative disorders such as Alzheimer’s disease (AD). RAS inhibition (RASi) drugs that can cross into the central nervous system have yielded unclear results in improving cognition in AD patients. Consequently, only one RASi therapy is under consideration in clinical trials to modify AD. Moreover, the role of non-genetic factors such as hypercholesterolemia in the pathophysiology of AD remains largely uncharacterized, even when evidence exists that it can lead to alteration of the RAS and cognition in animal models. Here we revise the evidence for the function of the brain RAS in cognition and AD pathogenesis and summarize the evidence that links it to hypercholesterolemia and other risk factors. We review existent medications for RASi therapy and show research on novel drugs, including small molecules and nanodelivery strategies that can target the brain RAS with potential high specificity. We hope that further research into the brain RAS function and modulation will lead to innovative therapies that can finally improve AD neurodegeneration.

## 1. The Renin–Angiotensin System (RAS) in the Brain

Since it was first described 120 years ago, new components of the renin-angiotensin system (RAS) in diverse tissues and physiological states are continuously discovered, teaching us about its inherent complexity [1], which becomes heightened in the brain, where the intricate distribution between the neuroglial network and vasculature, fluid homeostasis and metabolic circadian control has made it difficult to isolate and study. While there is a heated debate about the relevance of the RAS in the brain regarding control of cardiovascular function and blood pressure (BP) regulation [2,3], the most important feature of the brain RAS system is overlooked, and that is its role in cognition and neurodegeneration. To develop this important role for the brain RAS system, we will review some of the evidence supporting the existence of a brain RAS system and then discuss its relevance in cognition and concerning neurodegenerative diseases as Alzheimer’s Disease (AD). In addition, in the past year, the peripheral RAS was found to be involved in the pathogenesis of COVID-19, as SARS-CoV2 uses and modulates the expression of the angiotensin enzyme 2 (ACE2) to mediate its entry to the mucosa [4]. There were reports of neurological symptoms, and a recent imaging study from the UK Biobank (pre-peer reviewed) demonstrated volume loss in the left parahippocampal, orbitofrontal and insular regions, even after mild COVID-19 infection, showing that the central nervous system is also affected by COVID-19. These neurologic effects may involve the RAS [5], especially since ACE2 is expressed in neurons and microglia (Figure 1 and Figure 2).

## 2. All Key Players of the RAS Are Present in the Brain

There is decades-long evidence of the presence of renin and renin-like activity in neuronal cells and the brain in general. Renin was identified in primary neuronal and glial cells from rats [7] and immunohistochemical methods show the renin-like activity in rat and mouse brains [8]. The rise of molecular methods show that the promoter regions of several RAS genes are active in the brain [9], but nowadays, the expression of RAS genes can be consulted in single-cell sequencing libraries from mice and human brains, where it is evident that many RAS system components such as renin, angiotensinogen, aminopeptidases, and RAS-specific second messengers are expressed in one or several brain cell types, as shown in Figure 1 and Figure 2 [6,10,11,12]. This confirms seminal findings where angiotensinogen mRNA expression was first described in astrocytes [13] and later described in neurons as well [14]. Finally, there is extensive physiological evidence for the function of the brain RAS system and its effects, for example, when renin or angiotensin II (Ang II) is administered centrally in the brain of rats, producing an increase in systemic blood pressure (BP) [15], or with the use of transgenic rats that are deficient in brain angiotensinogen, which develop diabetes insipidus with systemic BP reduction [16].

## 3. Overview of RAS Signalling

The classical pathway for RAS modulation of blood pressure in the periphery starts with the release of renin by renal arterioles into the bloodstream. Renin then converts angiotensinogen (Agt) to angiotensin I (Ang I), which is then transformed to angiotensin II (Ang II) by the angiotensin-converting enzyme (ACE). Ang II binding to angiotensin 1 receptors (AT1Rs) results in vasoconstriction, while binding to angiotensin 2 receptors (AT2Rs) induces vasodilation [17]. In the context of the brain, both renin and angiotensinogen would be produced by astrocytes and other cells in regions such as the amygdala, reticular formation, CA1 and CA3 regions of the hippocampus [9]. Then, the BP modulation activity depends on endothelial cells and smooth muscle cells expressing ATRs and the G-protein coupled MAS receptors (MasR), which can induce vasoconstriction or vasodilation. We constructed an overview of the brain RAS signaling in different cell types based on single-cell expression data, shown in Figure 2. In this framework, generation of Ang II and AngIII is generally considered detrimental for neuronal function, given that binding to AT1Rs leads to vasoconstriction, which could prime protein aggregation and decrease glucose availability for neurons, as will be discussed later in this work. In the same framework, binding of Ang 1–7 to MasRs in glial cells often leads to vasodilation and anti-inflammatory effects [18]. In Figure 2, we assume pathways that are detrimental for neuronal function (marked in red) will be enhanced in AD, while neuroprotective pathways (arrows in blue) will be decreased. We want to emphasize that our proposed framework is based on the expression profile of single cells in the brain. Thus, neuron-glia interactions, modulation, and activity of brain RAS genes remain to be fully elucidated. In the same way, the absence of certain genes can suggest these would be expressed under different physiological or pathological conditions, such as inflammation or neurodegeneration.

## 4. Relevance of the Brain RAS in Cognition and Alzheimer’s Disease

High BP is a known risk factor for AD [19,20]. High BP may contribute to the development of AD and other types of dementia by negatively affecting brain perfusion, blood-brain barrier permeability and amyloid-beta vascular clearance. The effects of the RAS in the brain’s blood flow dynamics cannot be overlooked. Metabolic theories for neurodegeneration support that reduced blood flow to the brain with decreased glucose delivery to neurons are a cause of cell death and can worsen neuroinflammation [21,22,23]. Moreover, a cardiovascular risk factor (CVD) as hypercholesterolemia can contribute to this effect through the alteration of aminopeptidases A and N (AP-A and AP-N), as previously reported on a transgenic mouse model (CYP27Tg), overproducing 27-hydroxycholesterol (27-OH), a cholesterol metabolite that can cross the blood-brain barrier (BBB). In this model, glucose uptake is decreased in all major brain areas because of the increased levels of 27-OH. Together with the alterations in AP-A and AP-N, these animals showed reduced spatial memory in the Morris water maze at 9 months of age [24].

Importantly, recent genome-wide association studies have identified *ACE* as a gene linked to susceptibility for AD [25]. One of these ACE coding variants (ACE1 R1279Q) was recently investigated in novel mouse models and found to lead to hippocampal neurodegeneration and inflammation. This neurodegeneration was more pronounced in female mice, suggesting a possible mechanism for higher susceptibility for AD in women [26].

ACE expression in AD brain tissue was associated with amyloid-beta (Aβ) load and AD severity. Cerebrospinal fluid (CSF) levels of ACE were associated with Aβ levels [27] and risk for late-onset AD [28]. CSF ACE levels were elevated in mild cognitive impairment (MCI) and AD cases [29] and a positive correlation of cholesterol and 27-OH was found with RAS system actors in AD patients, linking cholesterol metabolism to brain RAS regulation. In another study, ACE activity was elevated in AD brains and correlated with Braak stages, while ACE levels were found to decrease in CSF from AD patients [30]. Furthermore, analyses from brain autopsies revealed increased levels of Agt and AngI/II in AD patients. The increase of Agt was found mainly in glial cells [29], suggesting a possible disruption of Agt synthesis or cleavage in the late stages of AD. Ang II mediates several neuropathological processes in AD [31] and it was recently targeted for intervention in phase II clinical trials of AD [32]. Recent evidence has suggested that angiotensin IV (AngIV) and its receptor (AT4R) can be potential therapeutic targets [33,34], since interventions on its signaling improved cognition and vascular flow in the brain of Alzheimer’s disease mouse models [35,36]. Thus, we have highlighted the generation of AngIV and its binding to AT4Rs as a beneficial pathway (Figure 2). Nevertheless, these results remain to be translated to humans and the broad activity and localization of AT4Rs make it a difficult pharmacological target.

BP modulation in the brain is also directly connected to the recently described glymphatic system. The glymphatic system is a complex network formed by the space between the vasculature in the brain and the end processes of astroglia [37]. This system oversees clearing toxic metabolites produced by neuronal activity and it is controlled by the circadian rhythm. During sleep, CSF flow is increased in the perivascular space and toxic proteins such as amyloid-beta are cleared at an increased rate compared to awake states [38]. Interestingly, hypertension decreases CSF flow through the glymphatic system due to its relationship with the arterial entry of blood to the brain [39]. It has been proposed that alterations in the glymphatic system function can prime protein aggregation and contribute to neurodegeneration in different proteinopathies, including AD [39]. Since the glymphatic system is difficult to study, its response to RAS inhibition (RASi) therapy, systemically or centrally, has not been reported to our knowledge, however, alterations in the glymphatic system are evident in spontaneously hypertensive rats [39], with implications for the impaired clearance of Aβ from the brain. Therefore, RAS modulation in relationship with CSF clearing through the glymphatic system of the brain is a very novel, and potentially promising line of research.

## 5. RAS Medications in AD

Due to the relationship between CVD risk and AD, the pharmacological regulation of systemic RAS was studied in relation to cognition and AD. During the last years, repurposing of antihypertensive drugs for AD therapy has been taken more and more into account [40]. RASi therapy was associated in several epidemiological studies with delayed progression of cognitive decline [20] and with reduced risk of dementia incidence [41,42]. Additional evidence showed a correlation between cognitive decline in aging and hypertension detected during midlife, among untreated hypertensive participants [43]. Nevertheless, other studies found no clear associations between RASi therapy and cognitive decline [44,45,46,47]. A meta-analysis found that RASi therapy did not significantly improve cognition [48]. However, two observational studies reported cognitive improvement and improved blood flow parameters with angiotensin receptor blockers (ARBs) as RASi therapy [49,50]. Angiotensin receptor blockers (ARBs) have been associated with reduced amyloid retention in patients in neuroimaging studies [51] and with less AD pathology in a post-mortem study [52]. Ramipril did not change Aβ levels in CSF [53], whereas ARBs showed a significant reduction of tau and p-tau among patients with mild cognitive impairment (MCI) [54] and reduced age-related decrease of CSF Aβ-42 in healthy patients after 24 months of treatment compared to other antihypertensive drugs [51].

Several studies on RAS medications in mouse models are helping to elucidate the mechanisms and roles of RAS in brain and AD, which we describe in the section below and summarize in Table 1.

## 6. RAS Medication in AD Mouse Models

### 6.1. ACE Inhibitors

In the last decade, several studies were performed in mouse models of dementia to investigate whether antihypertensive drugs targeting the RAS system could exert beneficial effects on cognition and the mechanisms behind that (See Table 1). Oral administration of the centrally active ACE inhibitor perindopril prevented and improved cognitive impairment in AD mouse models via inhibition of brain ACE activity [55,56]. Captopril, another BBB-permeant ACE inhibitor, was also reported to delay the development of neurodegeneration symptoms in aged Tg2576 mice by reducing hippocampal ACE activity and related ROS production [57]. Conversely, the same studies cited above [55,56] reported that non-brain-penetrating ACE inhibitors enalapril and imidapril did not significantly affect Aβ-induced cognitive deficits. In vitro studies found that expression of ACE promotes Aβ40 and Aβ42 clearance and that ACE inhibition by anti-hypertensive drugs can enhance Aβ deposition (Table 1) [65,66,67]. In contrast with these in vitro observations, most of the in vivo findings do not support the physiological role of ACE in the regulation of brain Aβ protein levels. Indeed, ACE-deficient mice did not show alteration in Aβ concentration [68] and inhibition of ACE by perindopril and captopril did not appear to affect cerebral Aβ accumulation and plaque distribution in AD mouse models [55,56,69]. The potential neuroprotective effect of BBB-crossing ACE inhibitors observed in mice might, therefore, be attributed to the reduction of inflammation and oxidative stress [56] induced by Aβ rather than a change in its levels or plaque formation. On the other hand, a more recent study on APP transgenic mice found instead that inhibition of ACE by captopril significantly enhanced Aβ deposition and that ACE-deficient mice had elevated Aβ42/Aβ40 ratio [70]. Pharmacological activation of ACE2 in symptomatic Tg2576 mice was found to reduce Aβ42 and IL1-β levels in the hippocampus and protect from cognitive decline, suggesting that ACE2 expression may exert a positive function in Aβ-related cognitive disorders [71]. Moreover, Kehoe et al. previously found ACE2 activity to be reduced in human AD brains [72]. In contrast with this, a more recent study reported that ACE2 was upregulated in the brain of AD patients [73]. These controversial observations suggest that further research is needed to elucidate the correlation between ACE2 gene expression and its enzymatic activity in the RAS axis. Thus, there are controversial results about the role of ACE in Aβ deposition in the brain, and further investigation seems to be necessary to identify the targets mediating the beneficial effects related to ACE modulation.

### 6.2. ACE Inhibitors in Aged Mice

The beneficial effects of ACE inhibitors have also been related to frailty and physical function during aging, going beyond cognition. A recent study on aged male and female wild-type mice was longitudinally performed to assess the effect of enalapril on frailty [74]. Chronic treatment with enalapril attenuated frailty in female mice more than in older male mice, without effects on blood pressure. Moreover, enalapril treatment resulted in a reduction in serum pro-inflammatory cytokines levels with higher beneficial effects in females than males, compared with control animals [74]. These sex-specific and systemic anti-inflammatory effects exerted by a non-BBB permeable ACEi may play a role in ACEi positive outcomes in higher brain functions. To further complicate matters, the brain permeability of many RAS drugs is not established, particularly in regards to chronic use or in older patients with higher BBB permeability [40].

### 6.3. Angiotensin Receptor Blockers

Angiotensin receptor blockers (ARBs), and particularly angiotensin II type 1 receptor (AT1R) antagonists, have also been shown to mediate positive effects on cognition in animal models of AD. There are extensive studies reporting evidence that AT1R blockers, such as losartan, valsartan, telmisartan and olmesartan can rescue or ameliorate cognitive impairment in AD mice [59,60,61,62,75]. Still, the mechanism related to the neuroprotective effects of ARBs remains unclear. Such as ACE inhibitor drugs, there are conflicting data on whether the positive effects of ARBs on cognitive functions are mediated by alteration of amyloid pathology or not. Valsartan showed the ability to decrease the levels of brain Aβ in primary cultures of cortico-hippocampal neurons [61]. Conversely, losartan did not change either different Aβ-species amounts or plaque load in APP transgenic mice [59,60]. Instead, losartan was reported to significantly reduce oxidative stress markers in the cortex and hippocampus of AD mice to wild-type levels [75]. Other angiotensin receptor subtypes were studied to elucidate the mechanisms underlying AT1R antagonism benefits. Blockade of angiotensin II type 2 receptors (AT2Rs) results in the abolishment of the neuroprotective events produced by AT1R blockers, indicating AT2Rs as potential contributors to some of the benefits induced by ARBs. Despite this, direct activation of AT2Rs failed to rescue AD-related symptoms and neuropathology in mice [75]. This suggests that AT2Rs play a role in the effects induced by ARBs, although AT2Rs agonism alone may not be sufficient as a candidate treatment to restore AD cognitive impairments. Angiotensin IV receptor (AT4R) function showed to be necessary to maintain losartan´s capacity of rescuing spatial learning and memory in young APP animals [35], further suggesting the implication of different angiotensin/angiotensin receptor cascades. A recent study in the same APP mouse model found indeed that angiotensin IV administration was capable of restoring Aβ-related cognitive impairments, together with a reduction of oxidative stress, independently of Aβ pathology [36]. This observation was supported by an increase in cellular proliferation, newborn cell number and dendritic arborization of hippocampal neurons in AD mice treated with angiotensin IV. The cognitive amelioration was also accompanied by restored cerebrovascular function. These findings propose ARBs, and, in particular, angiotensin IV/AT4R cascade components, as promising therapeutic targets for the prevention and treatment of AD-related neuronal and vascular deficits [33,34].

### 6.4. Modulation of Neuroinflammation by RAS Medications

Increasing evidence from recent years suggests a role of RAS in neuroinflammation associated with AD [76,77], proposing RASi medications as neuroprotective and, therefore, potential therapeutic agents in brain disorders. In the last decade, several studies in the mouse model of dementia and AD indicate the modulation of glial activation as one of the possible mechanisms mediating the positive effects of RAS inhibitors on cognition. The ACE inhibitors perindopril and captopril may prevent the activation of microglia and astrocytes in the hippocampus and cortex of mouse models of AD [56,58]. A similar reduction of microglia activation was also observed in a Parkinson’s disease mouse model after chronic captopril treatment [78]. The involvement of RAS on the activity of glial function within AD was further investigated upon ARBs administration. Telmisartan was shown to significantly decrease the production of pro-inflammatory mediators and ROS by murine microglial cells in vitro [63]. Moreover, the same study observed a reduction of hippocampal/cortical microglia and macrophages activation in vivo in 5XFAD mice. These findings propose a major role of microglia behind the beneficial effects induced by RASi in the brain. Centrally active blockers/inhibitors of RAS may, therefore, represent a promising treatment in addition to standard AD therapies such as cholinesterase inhibitors. In support of this view, inhibition of RAS was found to ameliorate cognitive disturbances by reducing microglia-related neuroinflammation also in other animal models of brain disorders, such as neuropsychiatric lupus and depression [79,80,81].

## 7. Novel RAS Drugs

Modulation of aminopeptidase-A (AP-A) and aminopeptidase-N (AP-N) in the brain has shown effective changes in systemic BP through intracerebroventricular injections in rats [82,83]. The mechanisms of BP regulation proposed in these works are either the production of AngIII [82] or the increased metabolism of it [83]. Another study found that aminopeptidase activity in the hippocampus could hydrolyze neuroprotective peptides such as enkephalin, showing that bestatin treatment is neuroprotective against CA1 neuronal death induced by oxygen-glucose deprivation [84]. In addition to its activity in the brain RAS, AP-A was recently involved in amyloid aggregation through N-terminal truncation of Aβ. In this study, inhibition of AP-A with RB150 (described below) restored the density of mushroom dendritic spines and reduced filopodia-like immature spines in hippocampal organotypic slices. Moreover, the work showed increased AP-A activity in early cases of AD [85]. As discussed earlier in this review, environmental risk factors for AD, such as cholesterol metabolites have been shown to alter AP-A and AP-N expression and correlated with decreased spatial memory in mice [24]. Thus, AP-A and AP-N represent pharmacological targets with proven efficacy in regulating blood pressure. However, the effect of their modulation in cognition and as a preventive strategy for AD has only begun to be characterized. To change AP-A and AP-N expression in the brain, diverse drugs are under development, and we will elaborate on a novel class of small molecules that are able to pass the BBB, and nanoparticle vectors that can also overcome the barrier problems of brain delivery.

### 7.1. Small Molecules for APA and APN Modulation. EC33 and its Prodrug RB150/Firibastat

EC33 ((S)-3-amino-4-mercapto-butyl sulfonic acid) is an orally administered AP-A inhibitor that cannot cross the BBB [86]. Nevertheless, when injected intracerebrally in the ventricles (up to 100 micrograms), EC33 inhibited brain AP-A activity in the 12 to 50 micrograms range in conscious mice [87] and, in a different study, it inhibited the production of AngIII as observed by radiolabelling of [3H]Ang III [88].

Since EC33 cannot enter the brain, the prodrug RB150, also known as firibastat, was developed. Firibastat can cross the BBB when administered orally and does not alter BP in normotensive rats. This prodrug can cross the BBB and enter the brain, where the cleavage of its central disulfide bridge by brain reductases releases two molecules of EC33. In rats, it showed BP reduction activity from 2 to 15 h post-administration [89]. As mentioned before in this work, the proven activity of firibastat for neurodegeneration is based on the inhibition of aminopeptidase activity over Aβ, which decreases the abundance of toxic Aβ species and its effect on neuronal physiology [85]. We have also mentioned that vasoconstriction and reduction of glymphatic flow could promote protein aggregation in the brain; therefore firibastat, through the activity of EC33 in the brain, could potentially improve these risk factors that influence cognition directly by lowering AP-A activity and AngIII levels. Firibastat reached a clinical trial phase IIb called NEW-HOPE (NCT03198793), in which it showed safety and efficacy BP lowering activity in high-risk populations [90].

### 7.2. NI929 and NI956/QGC006

EC33 was first designed as a systemic AP-A inhibitor and it showed that binding AP-A was sufficient to decrease its activity [86]. The nonpeptidic NI929 ([3S,4S]-3-amino-4-mercapto-6-phenyl-hexane-1-sulfonic acid), is a small molecule with potent activity as AP-A inhibitor, 10 times more potent than EC33 in vitro (K_i_=30 nmol) and able to cross the BBB into the brain [91]. When NI929 is dimerized by a disulfide bridge, it forms a dimer termed NI956, which major advantage is the ability to cross the BBB when administered orally. Moreover, NI956 would effectively downregulate AP-A activity without altering plasma sodium and potassium concentrations at a 10-fold fraction dose to that required for RB150. Unfortunately, this drug has only been tested in animals, and a clinical trial has not been reported.

### 7.3. Multistage Delivery Vectors (MDVs) and Nanoparticles for RASi Therapy

The development of MDVs obeys the need for a delivery system that can actively use the transporters in the BBB, such as transferrin [92,93]. It also follows the need to target a specific target or group of targets present in cells, which can also be used as cell-specific targeting [94]. MDVs were initially developed to treat certain types of tumors, which harbor cancer stem cells in their core and produce extensive layers of connective and vascular tissue [92]. In these tumors, MDVs have an initial coating that will allow the vector to enter the first layer(s) of tissue, which releases the vector coated with a ligand that provides target-specificity. After ligand-receptor binding, the vector is internalized into the cells and releases the drug, which exerts therapeutic effects. However, visualizing a tumor layer as the BBB gave rise to the idea of using the multistage coating to deliver directly into the brain [95,96].

MDVs no longer seems like a novel idea, however, recent discoveries on lysosomal signaling open the road for novel applications that can benefit from MDV delivery. In certain types of cancer, pain signaling is transduced by G coupled receptors (GPCRs), which upon activation are internalized into vesicles in nerve cells [97]. Normally, this mechanism leads to a decrease in the pain signaling due to the unavailability of the GPCRs to bind their ligands, however, in cancer, the internalized GPCRs can continue signaling from within the internalized lysosomes, which translates to chronic pain that does not respond to opioid treatment [98]. To tackle this problem, Jimenez-Vargas and co-workers designed a nanoparticle system that takes advantage of the acidic pH present inside the lysosomes where the GPCRs are internalized. These nanoparticles have a ligand that inactivates the G protein signaling downstream of the GPCRs and is only released under the lysosome acidic conditions, avoiding the non-specific binding associated with GPCRs inhibitors and decreasing the effective dose in orders of magnitude [99].

Angiotensin signaling depends, at least partially, on GPCRs that recruit arrestins to internalize angiotensin receptors as a way of desensitization [1,100]. The angiotensin receptor-arrestin complexes are directed towards the endosome, to assist in the recycling or degradation of the receptors [101]. Nevertheless, alternative signaling and scaffolding can occur after the formation of the ATR-arresting complexes, since arrestins can recruit several signaling molecules to the receptors already docked in endosomes [100]. This represents a possible signaling route for brain RAS that persist even after angiotensin peptides are not present, potentially hindering RASi therapies. Thus, it would be possible to direct angiotensin receptor blockers that will bind only under acidic endosomal conditions, allowing to regulate blood pressure signaling regardless of the amounts of Angiotensin isoforms present in the brain. This would mean alterations on the endosomal recycling of ATRs, since blocking them in the endosomes might enhance its degradation by the proteasome or autophagy mechanisms. Moreover, since the ATR blockers can be small molecules, they are suitable to be bound to MDVs and potentially be administered systemically [101,102]. We propose that the overall effect of blocking angiotensin signaling in the brain can act preventing glial activation (given their expression of only AT1Rs), and possibly vasoconstriction, improving blood flow and glucose availability for neuronal function. Proof of concept therapies in animal models showed that is possible to bind RAS inhibitors to polymer-based nanoparticles with BP-lowering effects [103]. In addition, lipid nanoparticles containing siRNA for angiotensinogen showed BP-lowering effects in rats [104], although this therapy is located upstream of the signaling pathway, while we propose to regulate the signaling after the ATRs have been activated, ensuring a more specific therapy that, for example, inhibit RAS signaling in astrocytes but not in vascular cells without altering global Ang levels.

## 8. Future Perspectives

From the studies mentioned here, several avenues for research in the brain RAS can be identified. We mentioned the use of nanoparticles to inhibit RAS signaling after ATRs activation, however, the mechanisms involving lysosomal signaling and scaffolding in the brain RAS remain largely unexplored. For example, AT1R activation leads to second messenger signaling that stimulates membrane proteases such as ADAM, which in turn can activate other tyrosine kinase receptors [105]. To our knowledge, it is unclear whether this phenomenon occurs in astrocytes or vascular cells in the brain.

24-S-hydroxycholesterol (24-OH) and 27-OH are cholesterol metabolites that can activate the RAS in neuronal cells in vitro [24,106]. Moreover, these oxysterols have clear modulatory effects on synaptic function, with CYP46A1 activation as neuroprotective [107,108,109,110,111] and high levels of 27-OH as detrimental [24,112,113,114]. When it comes to animal models for AD, there are very few examples that combine risk factors such as hypercholesterolemia with known genetic alterations leading to amyloidosis. CYP27Tg mice do not have neurodegeneration on their own, however, it is unknown how these phenotypes would synergize with genetic models that overproduce amyloid beta to promote neuronal death. On the contrary, CYP46A1 activation was studied as a pharmacological target for Alzheimer´s disease and Huntington´s disease [107,109,110], but the mechanisms offering neuroprotection are not yet well understood. CYP46Tg, a mouse model overexpressing CYP46A1 with high levels of 24-OH [115], has not been studied in the context of AD neurodegeneration, where it could promote neuroprotection and maintain cognition during aging, as suggested by behavioural studies on CYP46Tg alone [107].

Recent discoveries on the circadian modulation of the clearing system of the brain open the way to study new therapies on proteinopathies. Therapeutic approaches to treat amyloidosis have never considered that a higher clearance rate happens in the brain during sleep [116], which immediately suggests that some association could be found between melatonin and RASi therapies in patients. In rats, melatonin showed modulation of the insulin-regulated aminopeptidase (IRAP) in the pineal gland [117], and previously it was suggested that inhibition of IRAP can enhance cognition [118]. However, the relationship between these molecules remains to be studied in humans.

Finally, there is strong evidence that AD can be multifactorial [119,120,121,122], which highlights the importance of cohort stratification for study interventions in neurodegeneration. Age stratification improves the analysis of cohorts to estimate the risk of the APOE genotype [123]. In addition, correct patient-specific profiling of inflammatory biomarkers was studied as a strategy to improve diagnostics and prognosis in AD and Parkinson´s disease [124]. Therefore, studies looking at the use of RASi therapies in humans need to be reinforced with adequate and relevant patient stratification to find more clear associations between the brain RAS function and cognition. This will most surely lead to the discovery of better targets to improve cognitive function through modulation of the brain RAS and increase the alternatives for treatment in AD and other neurodegenerative diseases.

## Figures and Tables

**Figure 1 ijms-22-10139-f001:**
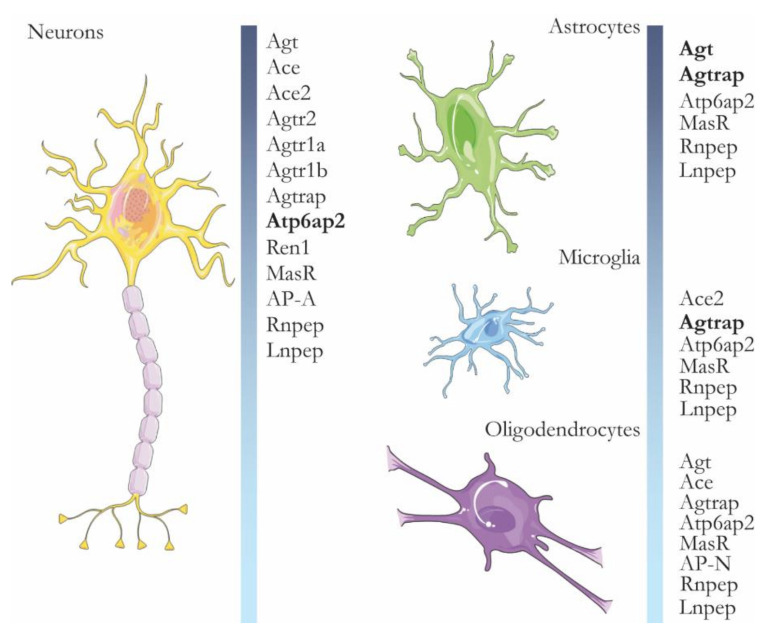
Genes of the RAS system in mouse brain cells. The figure shows genes expressed in the main cell types of the mouse brain, obtained by single-cell sequencing. Genes in black show increased expression (both copy number and number of expressing single cells) in that cell type. Agt-Angiotensinogen. Ace-Angiotensin-converting enzyme. Ace2 Angiotensin-converting enzyme 2. Agtr2-Type-2 angiotensin II receptor. Agtr1a-Type-1A angiotensin II receptor. Agtr1b-Type-1B angiotensin II receptor. Agtrap-Type-1 angiotensin II receptor-associated protein. Atp6ap2-Renin receptor. Ren1-Renin-1. MasR-Mas-related G-protein coupled receptor. AP-A-Aminopeptidase A (also known as Enpep). AP-N-Aminopeptidase N (also known as Anpep). Rnpep is the gene for aminopeptidase B (Also known as AP-B). Lnpep is the gene coding for the angiotensin IV receptor. Note that expression data were obtained from a WT animal and the lack of expression of genes in certain cell types could mean that such genes are expressed in a particular situation, such as inflammation or AD. Expression information extracted from http://linnarssonlab.org/cortex/ (accessed on 28 June 2021). Original article: [6].

**Figure 2 ijms-22-10139-f002:**
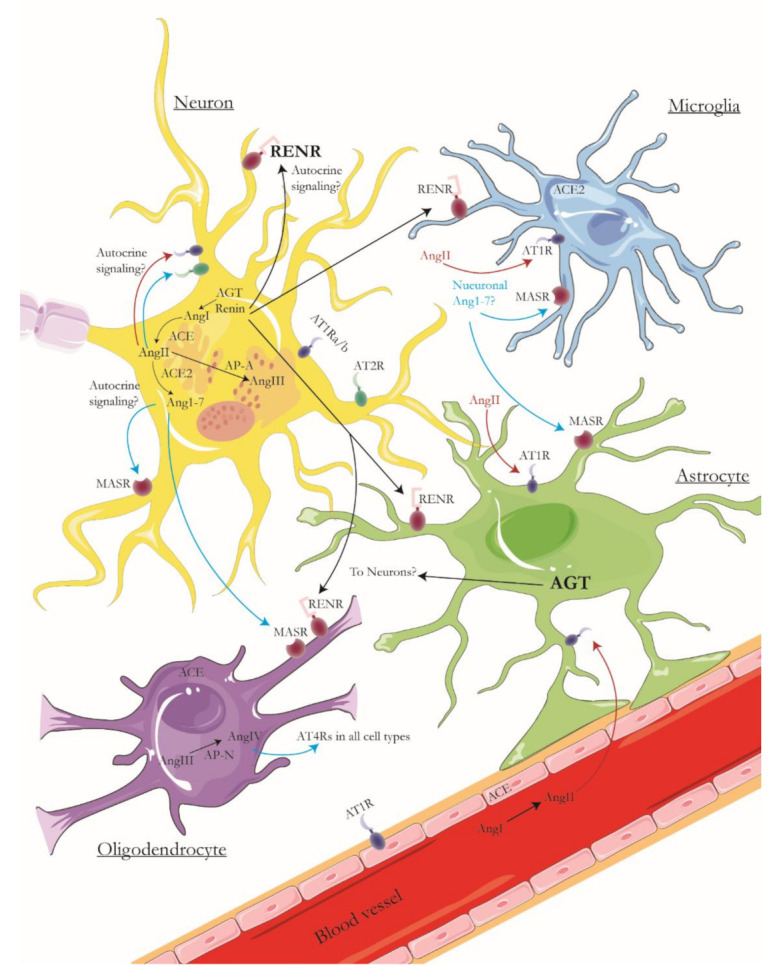
Components of the brain RAS. From the cell types in the brain, neurons are the ones that express the genes that comprise the classical angiotensin pathway, from the generation of angiotensinogen (AGT), which is converted by renin into Angiotensin I (AngI), which is then converted to Angiotensin II (AngII) by the angiotensin-converting enzyme (ACE). Renin can signal the renin receptors (RENR), for which neurons have a high expression level (marked in blacks). AngII can signal the receptors AT1R and AT2R, which are present in neurons. The binding of AngII to AT1Rs is usually considered detrimental (red arrows), while binding to AT2R has neuroprotective effects (blue arrows). AngII can be converted to Ang1-7 by the angiotensin-converting enzyme 2 (ACE2), and its binding to the Mas receptor (MASR) is often related to vasodilation, anti-inflammatory effects, and reduction of oxidation (also blue arrows). For the novel RAS pathways, aminopeptidases A and B (AP-A/AP-B) can convert AngII into angiotensin III (AngIII), which is then converted to angiotensin IV (AngIV) by aminopeptidase B. It is noteworthy that the next aminopeptidase in the pathway, aminopeptidase-N (AP-N) is only expressed in a small subset of oligodendrocytes, which creates the question of whether AngIII can reach oligodendrocytes for conversion into AngIV, which would then bind AT4Rs located in all cell types. In the same way, oligodendrocytes and microglia express only ACE and ACE2, respectively, thus raising the question of the origin of its canonical substrates. In addition, astrocytes have high levels of AGT expression, which suggests they could transport it to neurons for metabolism. Finally, AngII can come as well from the periphery since endothelial cells express ACE. Likely, BBB disruption in AD (star shape) facilitates the infiltration of AngII into the brain and its binding to astrocytes and microglia, which only have AT1Rs and can activate inflammatory cascades while at the same time generating vasoconstriction. AD can enhance red arrows while reducing blue ones. The figure is based on single-cell expression data shown in Figure 1. Cell types are not to scale, and no organelle localization is intended in this diagram.

**Table 1 ijms-22-10139-t001:** RAS medications in mouse models studies.

Drug	Class	Brain-Permeant	Mouse Model	Main Study Findings	Proposed Mechanisms	REFS
*Perindopril*	ACE Inhibitor	Yes	Aβ-injected mice; PS2APP mice	- Prevention of cognitive impairments - Reversion of cognitive deficits (working and recognition memory)	- Inhibition of brain ACE activities but not peripheral. - Reduction of microglia/astrocyte activation and oxidative stress	[55,56]
*Captopril*	ACE Inhibitor	Yes	Tg2576 mice; 5XFAD mice (and BV2 microglial cells)	- Chronic captopril slowed down the development of neurodegeneration signs	- Reduction of hippocampal ACE activity and ROS production - Reduction of IL-10 release by microglia - Decreased Aβ burden	[57,58]
*Enalapril*	Ace Inhibitor	No	Aβ-injected mice	- No effect on cognition - Very low inhibition of brain ACE activity	-	[55,56]
*Imidapril*	ACE Inhibitor	No	Aβ-injected mice	- No effect on cognition - Very low inhibition of brain ACE activity	-	[55,56]
*Losartan*	AT1R blocker	Yes	A/T mice; J20 APP mice	- losartan failed to restore spatial learning and memory in adult A/T mice but improved cerebrovascular activity - losartan ameliorated cognitive deficits in adult and aged J20 APP mice	- Attenuation of astrogliosis and normalization of AT1 and AT4 receptor levels (APP mice)	[59,60]
*Valsartan*	AT1R blocker	Yes	Tg2576 mice	- preventive valsartan administration attenuated cognitive dysfunction (improved spatial learning)	- Reduction of soluble extracellular oligomeric Aβ peptides in the brain	[61]
*Telmisartan*	AT1R blocker	Yes	Aβ-injected ddY mice; 5XFAD mice	- pretreatment with telmisartan prevented cognitive decline	- PPAR-γ activation and reduced Aβ deposition - Reduced activation of microglia and release of pro-inflammatory mediators and ROS	[62,63]
*Olmesartan*	AT1R blocker	No	APP23 mice; Aβ-Injected mice	- Attenuation of cerebrovascular dysfunction in APP23 mice; no reduction of brain Aβ levels - Improvement of cognitive functions in Aβ-Injected mice	- Decreased oxidative stress and neuroinflammation in the brain	[64]

## Data Availability

Not applicable.

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
