# Peer review of "Brain Renin–Angiotensin System as Novel and Potential Therapeutic Target for Alzheimer’s Disease"

_ijms, 2021, doi:10.3390/ijms221810139_

Round 1

Reviewer 1 Report

Two suggestions for this revised manuscript.

  1. In Graphical Abstract: should add angiotensin IV/AT4R cascade, it can restore Aβ-related cognitive impairments, together with a reduction of oxidative stress, independently of Aβ pathology, and cerebrovascular function. ATRs can be changed to ATR1, ATR2, AT4R, etc.
  2. Agtr1a or Agtr1b has not been present in glia in figure 1, and what does Rnpep stand for (figure 1)? AP-B (in fig. 2) did not present in the neuron in figure 1.

Author Response

We have now addressed all changes suggested by the reviewer.

Thank you.

Reviewer 2 Report

The authors have been well response for reviewer’s comments as well as they can.

This manuscript acceptable for publication in International Journal of Molecular Sciences.

Author Response

Thank you.

This manuscript is a resubmission of an earlier submission. The following is a list of the peer review reports and author responses from that submission.

Round 1

Reviewer 1 Report

In this review, authors summarized the brain renin-angiotensin system in AD. However, the manuscript was too preliminary content and lack of references. It might not be suitable itself for publication in International Journal of Molecular Sciences. There are critical concerns needed to be further elucidated.

Major comments

  1. Many articles have already reported RAS system through various journals. Entirely, this review focused RASi therapy. Therefore, authors should re-construct contents for RAS or RASi using more recent reports and add reference papers.

  1. Figure 1 have been described “genes of the RAS system in the brain cells”, however this figure just listed up the expressed gene name for neuronal cells or non-neuronal cells with cartoon. This information can replace the table section, therefore Figure 1 is not necessary for this review.

  1. Instead of Figure 1, authors may add new figure 1 for cell type specific expressed gene interrelationship between neuronal cells and non-neuronal cells using cartoon. I think new figure can give easy to understand to many researchers for complicate various gene relation in the brain.

  1. Graphical abstract is excellent in this review article. To concrete the paper, authors should add main figure based on graphical abstract including detail signaling in normal and AD brain.

  1. Lane 176~179: Authors described that Ace2 expression may exert a positive function in Aß-related cognitive disorders. According to recent published research papers reported that Ace2 protein was highly expressed in the AD patient brain. Authors should discuss with this controversial.

Ding et al. 2021 IJMS, https://doi.org/10.3390/ijms22041687

Wang et al. 2021 Alzheimer’s & Dementia  https://doi.org/10.1002/alz.12296

Minor comments

  1. Please keep the IJMS journal text rules (Figure, Fig etc)

  1. Authors more focused on RASi therapy in this review. To differentiate with other RAS review papers, title modification should be need.

Reviewer 2 Report

Thank you for the opportunity to review this manuscript entitled ‘The relevance of the brain renin-angiotensin system in Alzheimer's disease’ by Loera-Valencia and colleagues. The number of patients with Alzheimer's disease is increasing due to the aging population in developed countries. Many research groups are studying this disease. However, not enough research devote to the relationship between research on Alzheimer's disease and abnormalities in the renin-angiotensin system. In connection, presented review is relevant at the present time. The manuscript is well written and logical and presents data suitable for publication in the International Journal of Molecular Sciences in its current form.

Reviewer 3 Report

This article aims to review the evidence for the effects of the brain RAS in cognition and AD and summarize the evidence that links it to hypercholesterolemia and other risk factors. Moreover, some RASi medicine and novel drugs including small molecules and nanodelivery strategies were reviewed. Overall, the article gives an interesting historical and scientific perspective in the field of brain RAS and inspires novel therapies in the treatment of AD-like neurodegeneration. For improving this manuscript, this review article needs to be modified with the following comments and suggestions, although the authors attempt to summarize the importance of RAS in cognition disorders such as AD.

  1. What is the net effect of RAS in the regulation of blood vessels? Is it vasodilation or vasoconstriction? If ATRs can induce vasodilation and vasoconstriction at the same time, the summation of the effects of small molecules or nanoparticles listed in this article really can improve cerebral blood flow or not?
  2. Figure 1 needs to be revised, ATRs/MasR can be changed to ATR1, ATR2, AT4R, MasR, etc., and figure out their specific effects in the vessel, glial activation, or pathogenesis of AD, respectively.
  3. Line 80 “severity 80”
  4. line 139 “[38]. Click or tap here to enter text.”
  5. Has ACEi any other roles in this model of Figure 1, except for the regulation of Amyloid-beta deposition?
  6. Several AT1R antagonists listed in the present article also have effects in regulating the levels of ROS, inflammation, and A-beta. These mechanisms should be displayed in Figure 1.
  7. The introduction of aminopeptidase-A (AP-A) and aminopeptidase-N (AP-N) needs to be improved, especially their relationship with brain insult.
  8. Please detailedly describe the possible effects of EC33 and its prodrug RB150/firibastat as well as NI929 and NI956/QGC006 in AD-like neurodegeneration.
  9. In the section “Angiotensin receptor blockers”, the authors mentioned the AT2R and IV/AT4R cascade that could be a promising therapeutic target for the prevention and treatment of AD, authors should more elaborate on the evidence and mechanisms that support these receptors as a candidate in the treatment of AD. 
  10. In the section “Novel RAS drugs”, the authors focus on modulation of aminopeptidase-A (AP-A) and aminopeptidase-N (AP-N) and mention that the effects of their modulation in cognition and as a preventive strategy for AD are completely uncharacterized, please provide more rationale for the possibility as a candidate in the treatment of AD except regulation of blood pressure.